# Agreement threshold on Axelrod's model of cultural dissemination

Pádraig MacCarron[1,2]*, Paul J. Maher[1], Susan Fennell[2], Kevin Burke[2], James P. Gleeson[2], Kevin Durrheim[3], Michael Quayle[1,3]

**1** Centre for Social Issues Research, University of Limerick, Limerick, Ireland, **2** Department of Mathematics & Statistics, MACSI (Mathematics Applications Consortium for Science and Industry), University of Limerick, Limerick, Ireland, **3** University of KwaZulu-Natal, Pietermaritzburg, South Africa

* padraig.maccarron@ul.ie

**Data Availability Statement:** The data is generated randomly using the model presented in the paper.

**Funding:** PMC, PM and MQ are funded by the European Research Council (ERC) under the

## Abstract

Shared opinions are an important feature in the formation of social groups. In this paper, we use the Axelrod model of cultural dissemination to represent opinion-based groups. In the Axelrod model, each agent has a set of features which each holds one of a set of nominally related traits. Survey data has a similar structure, where each participant answers each of a set of items with responses from a fixed list. We present an alternative method of displaying the Axelrod model by representing it as a bipartite graph, i.e., participants and their responses as separate nodes. This allows us to see which feature-trait combinations are selected in the final state. This visualisation is particularly useful when representing survey data as it illustrates the co-evolution of attitudes and opinion-based groups in Axelrod's model of cultural diffusion. We also present a modification to the Axelrod model. A standard finding of the Axelrod model with many features is for all agents to fully agree in one cluster. We introduce an agreement threshold and allow nodes to interact only with those neighbours who are within this threshold (i.e., those with similar opinions) rather than those with any opinion. This method reliably yields a large number of clusters for small agreement thresholds and, importantly, does not limit to single cluster when the number of features grows large. This potentially provides a method for modelling opinion-based groups where as opinions are added, the number of clusters increase.

## Introduction

To understand the societal structure of opinions, or attitudes, it is necessary to model the emergence of the groups that bind them. Much research demonstrates that people evaluate and understand their environment with reference to relevant social groups [1–3]. Importantly, shared opinions and beliefs are a defining feature of social groups [4, 5] and group identity can be fostered by coordinating attitudes. In particular, opinion-based groups are social structures in which people are connected by the opinions they share; and clusters of opinions become interlinked signifiers of group identity when they are jointly held by the members of a group [6].

European Union's Horizon 2020 research and innovation programme (grant agreement No. 802421: https://ec.europa.eu/programmes/horizon2020/en/h2020-section/european-research-council SF, KB and JPG are funded by the the Irish Research Council (SF) and from Science Foundation Ireland (JPG, grant numbers 16/IA/4470 and SFI/12/RC/2289P2: http://research.ie/ https://www.sfi.ie/ The funders had no role in study design, data collection and analysis, decision to publish, or preparation of the manuscript.

**Competing interests:** The authors have declared that no competing interests exist.

In this paper we demonstrate that an adapted version of Axelrod's model of cultural dissemination [7] can be used to model opinion-based groups. Axelrod modeled people as each holding one of several available *traits* for each of several *features*. The individual's *culture* is defined as their combination of traits; and multiple individuals are said to share the same culture if they are spatial neighbours who share the exact combination of traits. We remove this spatial constraint when identifying clusters and instead get groups of agents sharing the same traits for each feature, we refer to these as *opinion-based groups*.

The social structure of opinion-based groups has similar properties to Axelrod's model. Specifically, agents in Axelrod's model are linked according to the values they hold (which Axelrod called "traits") on a given number of cultural "features". Opinion-based groups are formed by people holding a particular selection of attitudes. In principle, conceptualising Axelrod's nominal cultural features and traits as ordinal attitudes will allow us to model the emergence of opinion-based groups with a data structure that maps cleanly on to raw survey data. Note that opinion surveys typically use ordinal Likert-type response items (e.g. an item with several response options from Strongly Disagree to Strongly Agree). While these are frequently treated as interval data in analyses (ie. assuming consistent intervals between scale points and allowing arithmetic operations such as addition and subtraction), there is a strong argument that individual Likert-type items should properly be treated as ordinal, allowing only non-arithmetic operations [8]. The original Axelrod model of cultural dissemination relies only on swapping, and our adaptions adds ranking, making it an excellent fit with Likert-type data.

The tendency toward consensus in the standard Axelrod model poses a further problem for modelling the emergence of opinion-based groups with attitudinal surveys. Specifically, it has been shown that for the final state to have many clusters, a high initial number of traits $q$ is required [9]. Given that attitudinal surveys naturally assess more attitudes than response-options, an Axelrod simulation of such data would converge towards uniformity. However, in real social systems survey data frequently reveals stable systems of cultural diversity [10]. Our adaptation of the Axelrod model is intended to account for diversity in the structure of opinion-based groups. The idea to use surveys with the Axelrod model is not new, for example, [11] use survey data to set the initial states of the agents. In this paper, however, we approach this as a theoretical model and do not use empirical data as in, for example, [12–14].

There have been many variations on the standard Axelrod model (some even proposed in the original paper) often attempting to promote multiculturality. Some examples of variations are the introduction of noise [15], changing the topology from a lattice to a complex network [16], the effect of media represented by an external field [17], and, more recently, modelling it on a multi-layer network where only interactions can occur between nodes on specific layers [18]. Another variation treats each feature as a continuous variable which pull closer together when they agree or repel when they disagree [19]. This is similar to the Deffuant model [20] with more than one feature.

Here, we take a similar approach to the idea of *bounded confidence*. In this type of model, an agent only takes account of a neighbour's opinion if it is within a certain interval of that [21]. We treat each specific feature as having a discrete value associated with it. We then only allow an interaction when the traits are within a certain distance from each other. This corresponds with research from social psychology which demonstrates that social judgements mediate attitude change [22]. For example, according to social judgment theory [21], the amount of attitude change caused by an interaction should depend on the perceived distance between the communication and the respondent's original view. Specifically, at any given time there is a range of positions each person is open to holding, known as the "latitude of acceptance". Importantly, this range is anchored by one's current opinion. Similarly, decision making research demonstrates that people tend to display bias in evaluating evidence by favouring

views that correspond to their existing attitudes [22]. For example, empirical evidence shows that people rate arguments as more compelling when they correspond to previously held attitudes [23]. In summary, it is likely that interaction does not always lead to assimilation and that social influence is anchored by existing opinions.

This is similar to the approach in [24] but on the traits instead of the features. In general, if the number of features are much higher than the number of traits, there will be consensus, however, bounded confidence has been suggested as a method for compensating for this [11]. Similarly, [25] show that adding influence between those with similar options is sufficient for cultural diversity. A model similar in spirit in [26] uses an open-mindedness parameter, if two clusters are similar then there is higher probability they will coalesce, there is also a probability a group will fragment. This open-mindedness parameter is similar to the bounded confidence threshold. A model, with only binary options for the traits inspired by the Axelrod model with bounded confidence can also lead to multiple clusters [27]. A key difference in the model we present to other approaches that use bounded confidence on the Axelrod model is we take this threshold into account when computing who an agent can interact with, and then again when seeing if they are converted. This only introduces a single parameter and does not restrict the number of traits.

In this paper, we firstly use bipartite network visualisations to identify opinion-based group structure in the standard Axelrod model. Secondly, we introduce an agreement threshold similar to the notion of *bounded confidence*. Axelrod's original interaction mechanism was based on the principle of homophily—"the likelihood that a given cultural feature will spread from one individual to another depends on how many other features they may already have in common." If the nominal traits available to each agent are ordered then there is another dimension for homophily. A small modification to the interaction rules allows us model the types of survey-based data evident in the field of opinion-based groups.

## Methods

### Standard Axelrod model

In the social sciences, attitudinal surveys are the most common way of measuring individual opinions. We wish to model participants holding attitudes and the groups they form as a result. To do this we start with with Axelrod's model of cultural diffusion [7]. Here, each agent takes a position on a set of features; and that position must be one of a fixed set of traits. We begin by visualising this model using a bipartite graph—a graph with two types of nodes where edges must connect nodes of different types. One type of node represents the agents and the other represents their trait on a specific feature.

The original Axelrod model on a lattice (without periodic boundary conditions) works as follows: Each individual has a feature set $\mathcal{F} = \{f_1, f_2, \ldots, f_F\}$ where $f_k$ is each specific feature. There are a total of $F = \dim(\mathcal{F})$ features and each feature $f_k$ has $q$ traits (initially assigned at random). At each time-step, individual $i$ and one of its neighbours $j$ are chosen at random. With a probability equal to the number of common features over the total number of features, person $i$ will copy one of the traits they do not have in common with $j$.

Fig 1 shows the Axelrod model for 100 nodes on a lattice with $F = 3$ and $q = 5$ paused after 40,000 time steps. The absence of a line represents all five features in common, dotted lines represent three or four features in common, dashed lines represent one or two shared features and a thick solid line represents no features in common. When multiple clusters are found, the lattice is divided into regions which are physically separated from each other as seen here. A cluster with no edges indicates that those nodes all have the exact feature configuration.

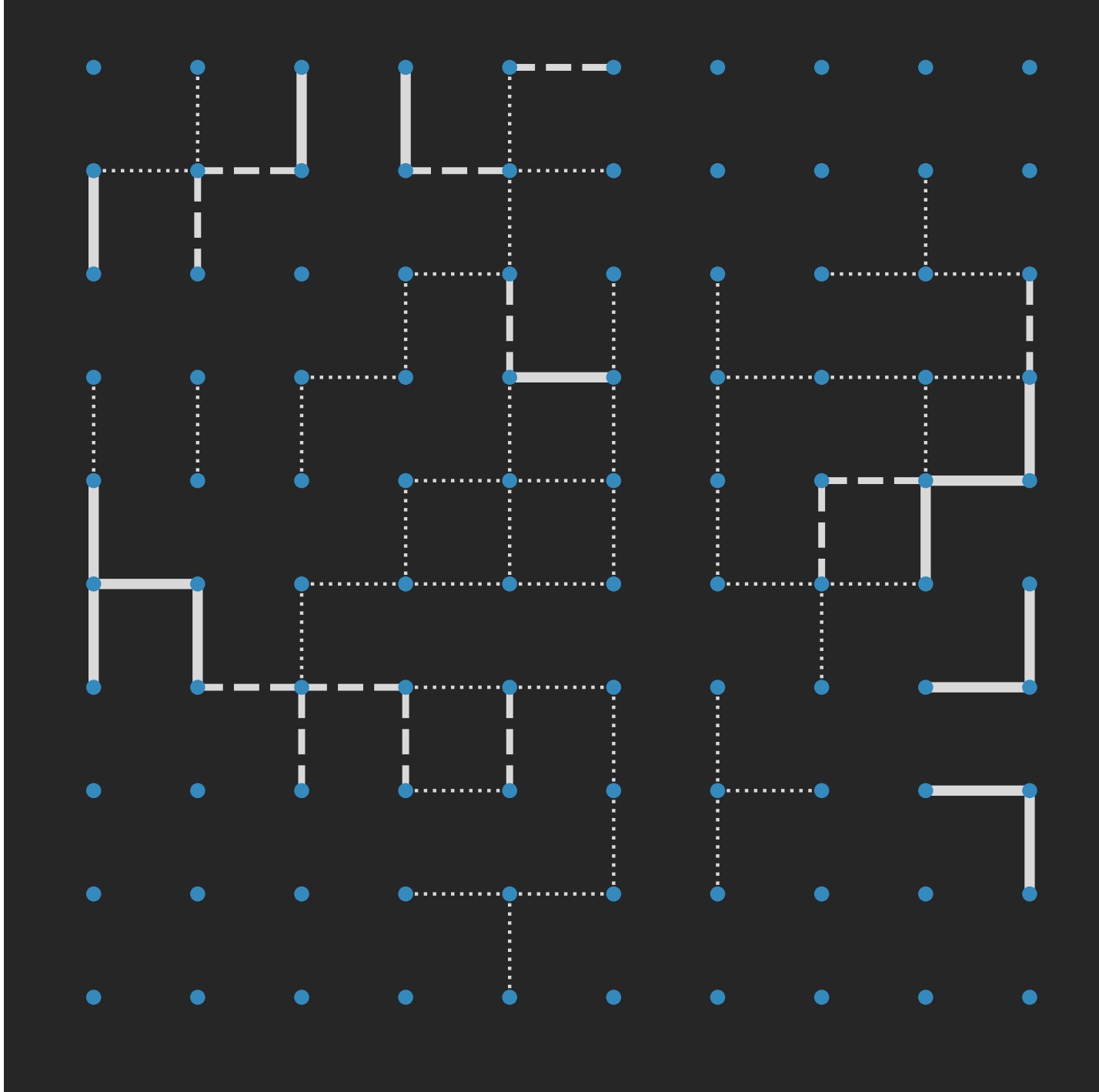

**Fig 1. An Axelrod simulation on a lattice with $N = 100$, $F = 3$ features and $q = 5$ traits paused after 40,000 time steps (i.e. not final state).**

A bipartite network representation of the same data is shown in Fig 2. Here, the underlying feature-trait combinations are revealed, with each feature displayed using a different colour. For example, we observe that features 1 and 2 both have a majority but feature 3 is split between $q = 1$ and $q = 4$. This is particularly useful if using this network to represent survey

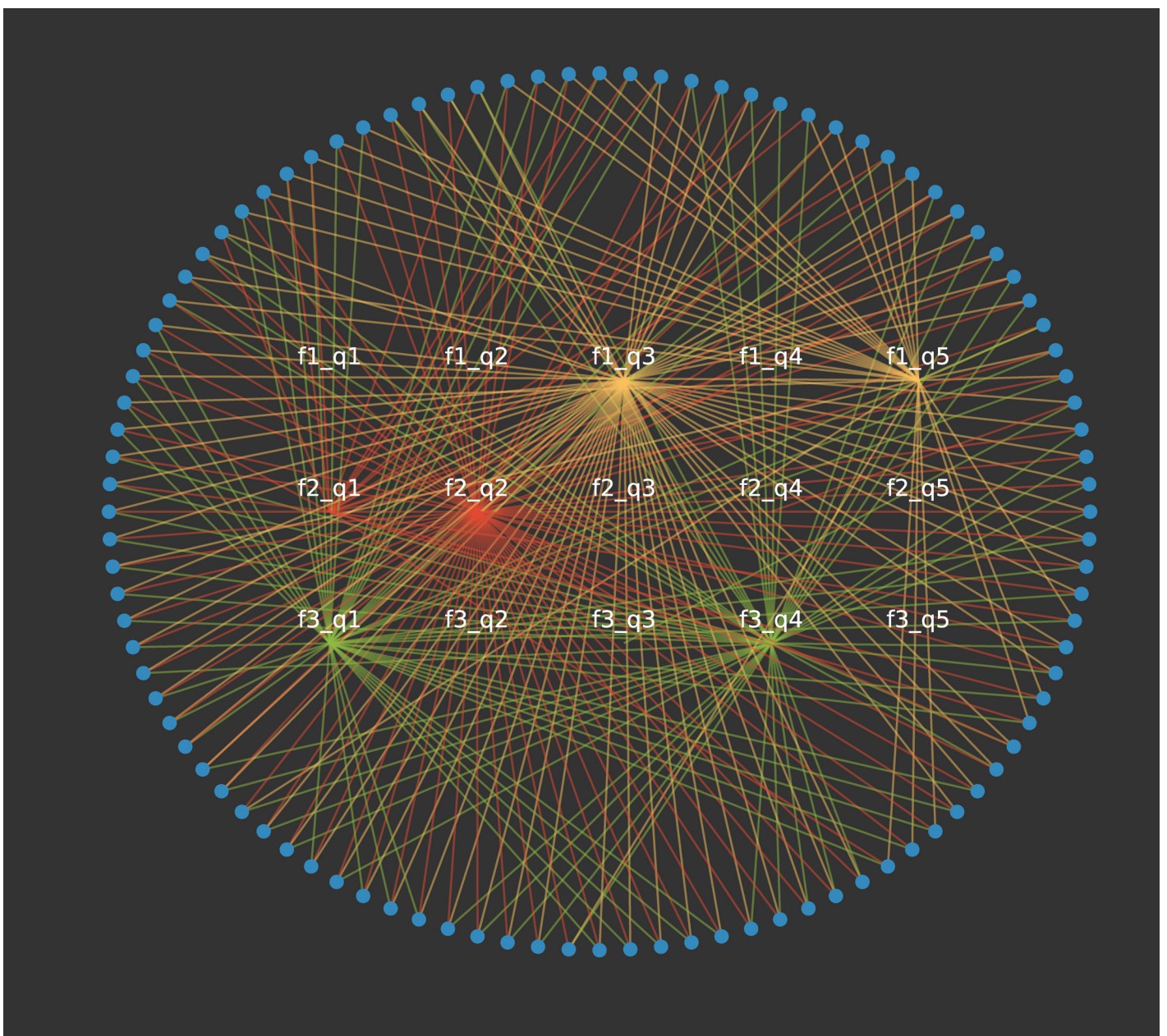

**Fig 2. The bipartite representation of the Axelrod simulation in Fig 1 after 40,000 time-steps with *F* = 3 features (different colours) and *q* = 5 traits displaying which feature-trait combinations are selected.**

data. If there are three questions with a five-scale response, it is clear which response is the most favoured per question.

Projections can be taken from this representation to show which feature-trait combinations are frequently held together (see for example Fig 4) and to show the opinion-based groups that form (see Fig 5). In the latter projection, if just full agreement edges are shown (i.e. where the agent share the same traits for each features), this gives the clusters usually discussed. If this is reduced to one minus the full agreement, it will display clusters who agree on all but one trait. An example of this can be seen in Fig 5 where different coloured edges represent different levels of agreement between the clusters.

## Agreement thresholds on the traits

One key difference between survey data and the Axelrod model is that in the Axelrod model the traits for a given feature are nominally related (i.e. not ordered) but in survey data the response options are related to one another, or on a scale (e.g., a Likert-type scale) [28]. We introduce a mechanism to allow for traits to be related to each other below.

A shortcoming of using the Axelrod model to represent survey data is a tendency toward uniform convergence in the final state as the number of features increase. Given that attitudes are negotiated properties of social interaction [29], it is plausible to assume that people hold a broad range of continuous positions on a given attitude. However, we are interested in how attitudes are communicated. People typically communicate attitudes verbally, which leads to a certain level of imprecision in their interpretation [30, 31]. We argue that by scaling opinions to a limited number of points on a continuous scale, surveys capture general limitations in attitude communication and interpretation. Furthermore, continuous variable models show a tendency for opinions to move towards the mid-point of the scale [32, 33] and thereby fail to capture the stubbornness of extremists [34].

In the standard Axelrod model above, if two neighbours interact, the node originally chosen copies the state of one of its neighbour's features that they do not already agree on. We argue that purely copying an attitude of an individual with whom you interact even if you strongly disagree with this attitude is unrealistic. Hence, we introduce an agreement threshold, $a$, analogous to a latitude of acceptance [29] to compute the interactions. The idea is that there exists a range of positions a person is willing to hold with this range being anchored by their current position within which they copy.

Instead of interacting with someone with probability proportional to the number of common features, we now calculate the probability to interact as the number of features within this agreement threshold over the total number of features, with interaction leading to a copying of one of those traits (i.e., only traits within the agreement threshold can be copied).

Once the state to copy is chosen, it only copies if the difference in values of the chosen feature between interacting individuals $i$ and $j$ is less than the threshold, $|f_{k,i} - f_{k,j}| \leq a$. For example, if the threshold is set to the smallest difference between response options in a survey scale, then only those who are one point away from each other on the survey scale will interact.

In contrast with standard Axelrod, here the probability for interaction between two nodes is increased when the agreement threshold is included as features which are $a$ traits away now add to the probability for interaction. However, when interaction occurs, the only features that can change are those within the agreement threshold. In standard Axelrod once interaction happens one feature will be copied, thereby increasing the probability of interaction in the future. In this version only the features which are already within the agreement threshold change, meaning the probability for interaction in the future is the same (unless a trait is changed by a different neighbour).

Fig 3 displays the final state for the lattice and bipartite visualisations of this model with $F = 6$, $q = 3$ with the lowest possible agreement threshold $a = 1$. For standard Axelrod, this would lead to consensus, however, here we see we have different regions and from the bipartite representation, half of the six features do not reach consensus. This is a simple adjustment to standard Axelrod that yields more than one cluster even where the number of features exceeds the number of traits per feature as they would in a typical survey.

## Projections

Visualising Axelrod on a bipartite space helps us realise that projections can be made to observe the connections between feature-trait combinations and nodes. A 1-mode projection

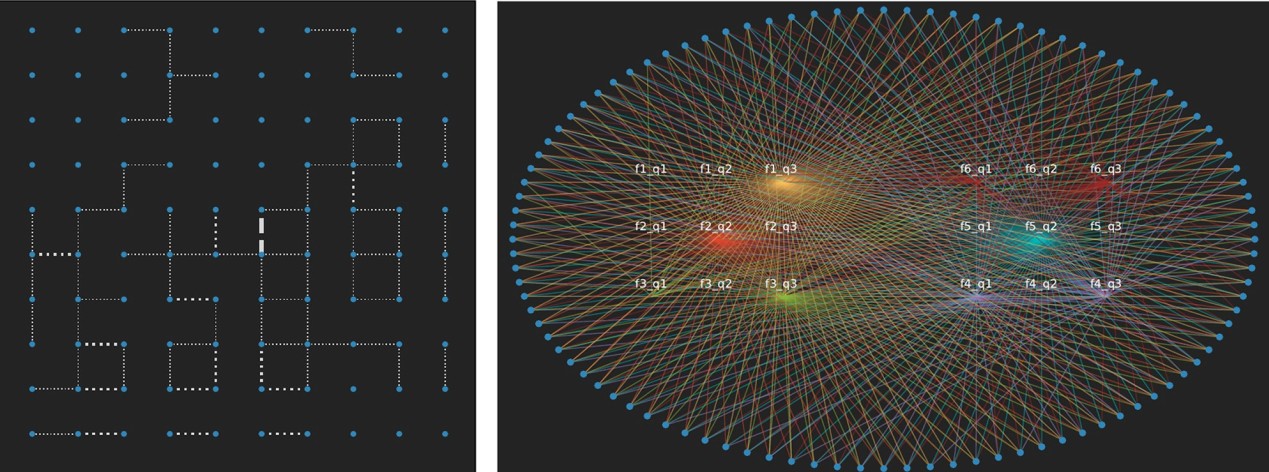

**Fig 3. The lattice and bipartite visualisation for the final state of Axelrod with the lowest agreement threshold on the traits for $F = 6$ and $q = 3$.**
Three of the features reach consensus in this realisation.

can be generated showing features as nodes with an edge indicating that two features are held concurrently by an agent, and therefore overall which attitudes are strongly connected.

The projection for $F = 6$, $q = 3$ is shown in Fig 4 where the weight of an edge is the number of pairs of nodes that share that feature-trait combination. These are split in thirds from weakest to strongest for visualisation purposes. The strongest edges are between the three attitudes where consensus is reached as these are held together most commonly.

Fig 5 shows the 1-mode projection for agents where agents are nodes, linked if they share an attitude in common. The colour of the edge represents how many attitudes they share, and we can identify a number of clusters where they share all six attitudes (with white edges). Note that this figure only displays the giant component. As the edges are created by shared links, if a node has no features in common with any other node, it will not be connected and therefore not appear in any clusters.

## Number of clusters

To count the number of clusters, we count the number of components where the nodes agree on all $F$ features in the final state. (Note that for larger systems, and larger agreement thresholds, it can make sense to count clusters that differ by one instead of assuming they all agree, however, here we are just taking the most simplified approach initially).

Fig 6 shows the average number of clusters for varying features for the agreement-threshold model with $q = 5$ (left) and $q = 7$ (right) for 1,000 runs with 100 nodes. For small $a$, as $F$ increases the average number of clusters increases as there is a much larger feature-trait space. Due to the large number of different opinion based groups, it is harder for large clusters to form so the cluster size gets smaller. As $a$ increases, there are more conversions and thus less clusters in the final state.

As Fig 6 focussed on the mean of 1,000 runs, we next display the density plot showing the distribution of the number of clusters for one feature in Fig 7. The left panel shows the change in the number of clusters in increasing $a$ with 4 features and 5 traits. In the right panel we increase to 8 features. Again we observe as $a$ increases the number of clusters reduces to as there are more conversions and larger groups. For low agreement thresholds however, there

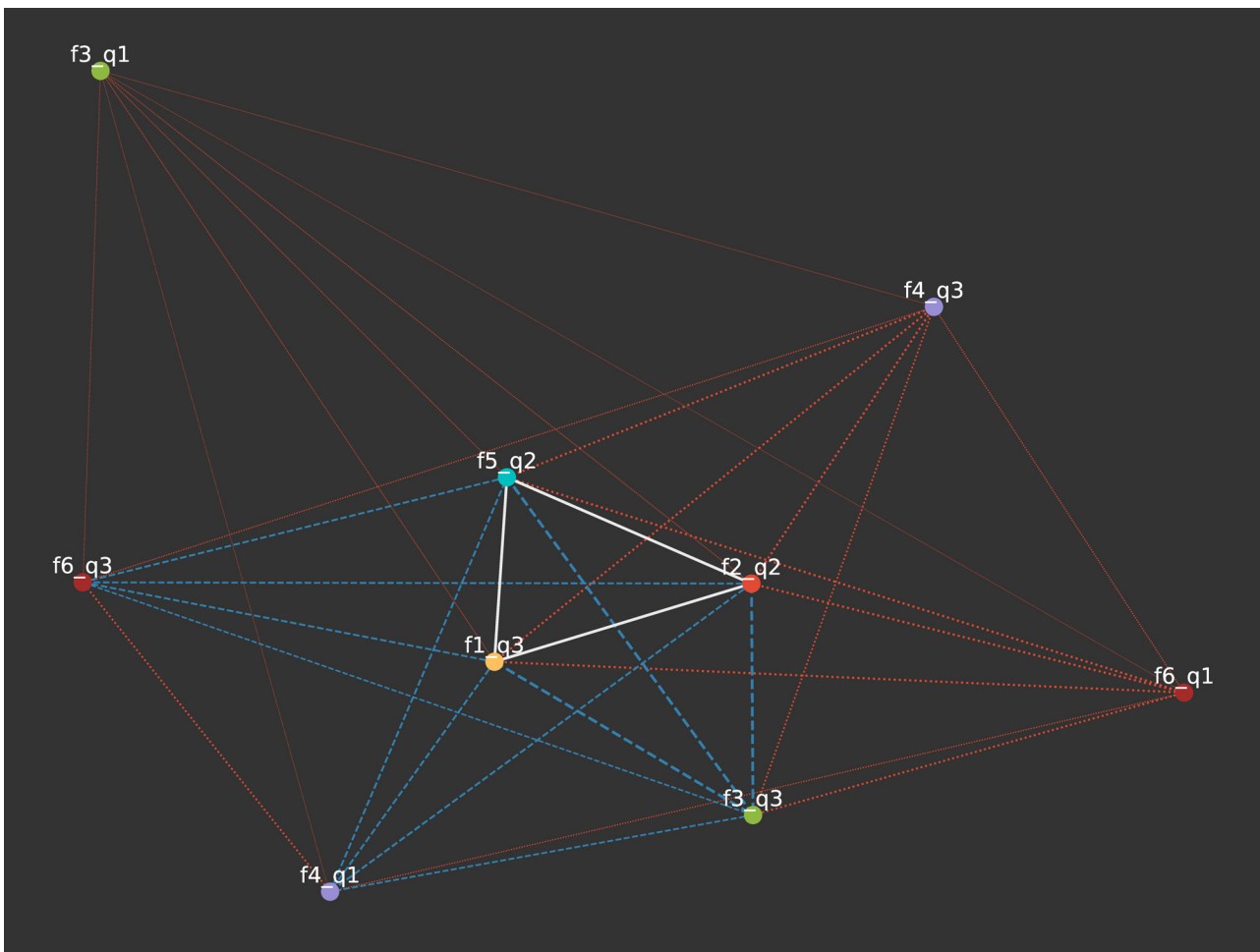

**Fig 4. The attitude-space from Fig 3.** White edges represent the top third strongest edge weighs, blue the middle third and red the weakest third of the edges. Here we see that the three attitudes for which consensus was reached above have the strongest edges as more nodes hold them in common.

are a much larger number of clusters and with large *F*, there are very few large clusters with many isolates who don't have the same feature-trait combination as any other nodes.

Finally, in Fig 8 we increase the system size to 1,024 with *q* = 7. In the left panel we observe the same change in mean number of clusters per feature as in Fig 6. In the right panel, we show the density plot, there is a larger jump between *a* = 1 and *a* = 2 when the system size increases.

Note that for agreement threshold *a* = *q* − 1, every interaction will lead to copying so this will quickly yield full consensus. Also note that in all these simulations, due to interactions now occurring when agents have a similar trait instead of exactly the same as in standard Axelrod, these tend to reach their final state faster than standard Axelrod.

## Conclusion

Here, we added an agreement threshold to the Axelrod model. This threshold yields many clusters for small values of *q*. This addition is an adjustment to standard Axelrod interaction rules that only allow those with similar features to influence each other. The resulting model is suited to natively modelling survey data, which frequently consists of Likert-type responses,

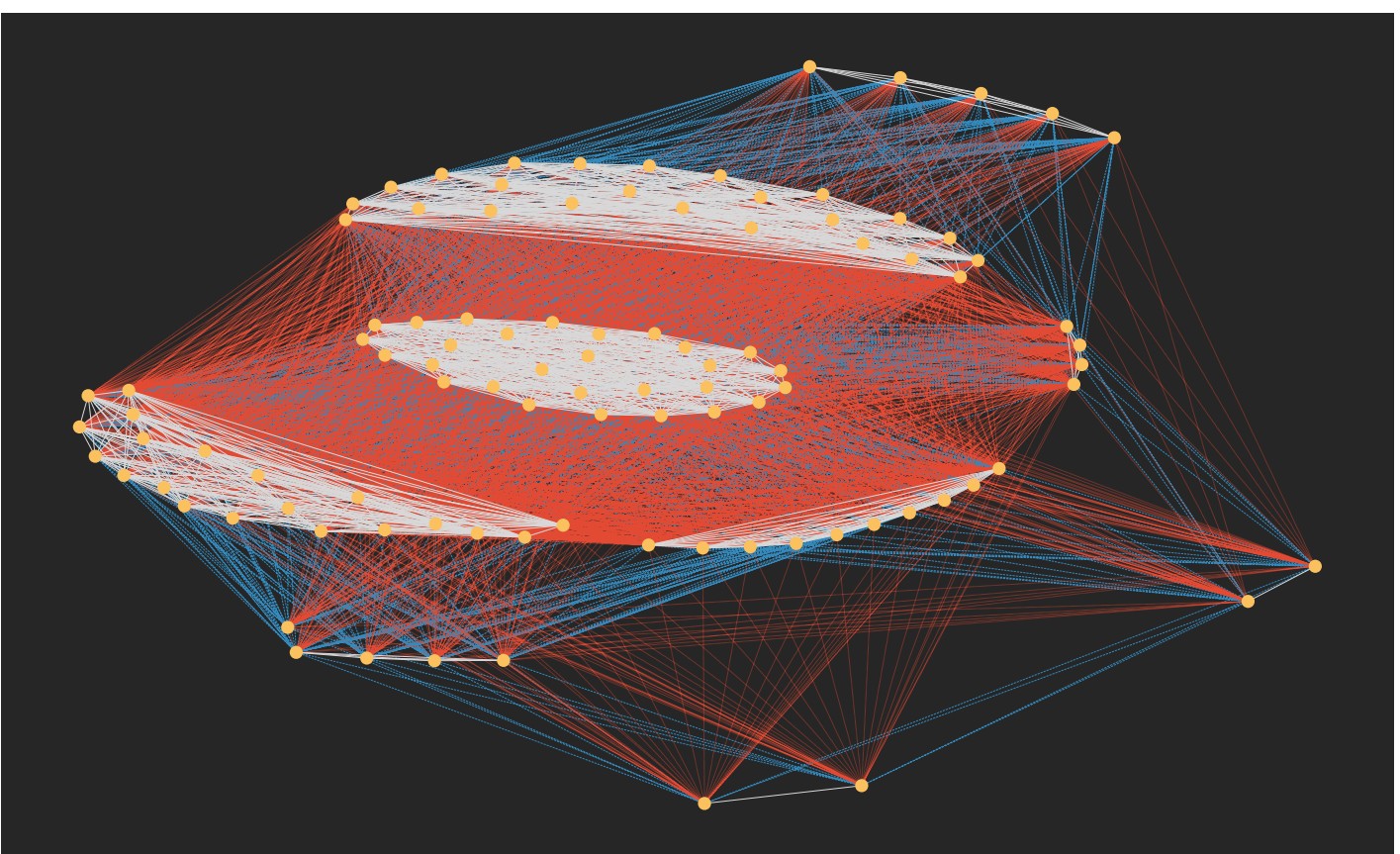

**Fig 5. The node-space projection (giant component) from Fig 3.** Here, two nodes are linked if they hold an attitude in common. White links represent nodes agreeing on all six attitudes, blue edges represent five common attitudes, and red represent three or four.

since only ranking and swapping operations are required thus respecting the ordinal nature of the data. The model also avoids the homogoneous end-state which tends to be the case when the standard Axelrod model is used on survey-type data, as there are almost always features than traits (i.e. more items than response options).

In this model, extremists are less likely to change their position than moderates. Due to the agreement threshold, an agent with an attitude on either extreme can only move in one direction, an agent in the middle can move in either direction so is twice as likely change their position. This helps with avoiding a problem some models have causing agents to move towards a centrist attitude and achieving polarisation [35]. This behaviour is also observed in real social systems [34].

Here, this model is entirely theoretical and uses random seeding of the initial system on a regular lattice. Going forward we would like to use empirical data to seed the initial values rather than randomly assign them, something which is used in the following approaches to Axelrod models [11–14].

Further work needs to be done on testing this method on topologies other than a lattice. Similarly, varying $q$ per feature should be tested as there is no reason the number of traits per feature should be the same. This last suggestion is made in [11]. A further extension could specify that if the agreement threshold is larger than one, the traits move towards each other rather than one agent copying the other. We also wish to relax the rules for group membership, here we took groups where everyone agrees with everyone, however, going forward we wish to

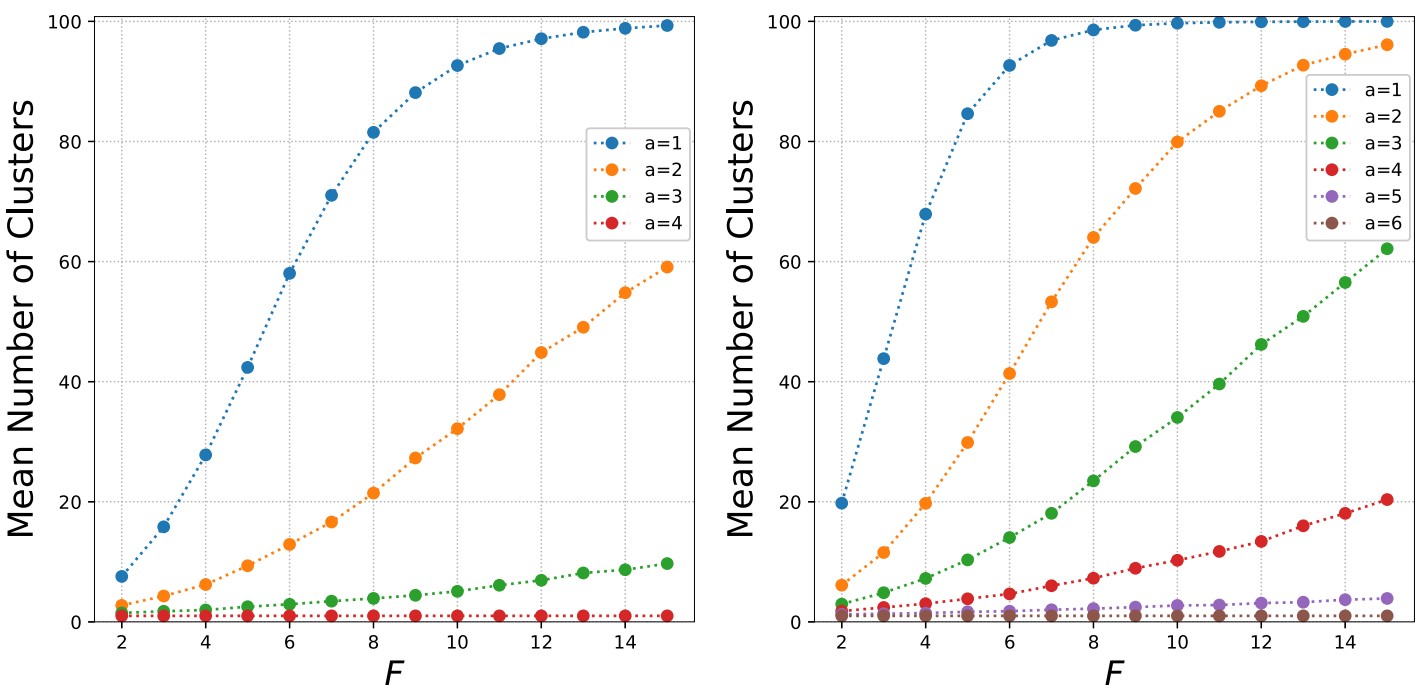

**Fig 6. The mean number of clusters per feature for *q* = 5 (left) and *q* = 7 (right) for different values of the agreement threshold *a*.** Each point is the mean number of clusters from 1,000 simulations. As the number of features increases, the number of clusters also increases but the size of the clusters decrease.

investigate this with agreement on a smaller number of core attitudes. In this case we would get larger clusters similar to when *F* is in the middle range of Fig 6.

Future work will seed this extended Axelrod model with real survey data to compare the sensitivity or resilience of various real-world attitude network topologies to cultural diffusion.

(a)                                                    (b)

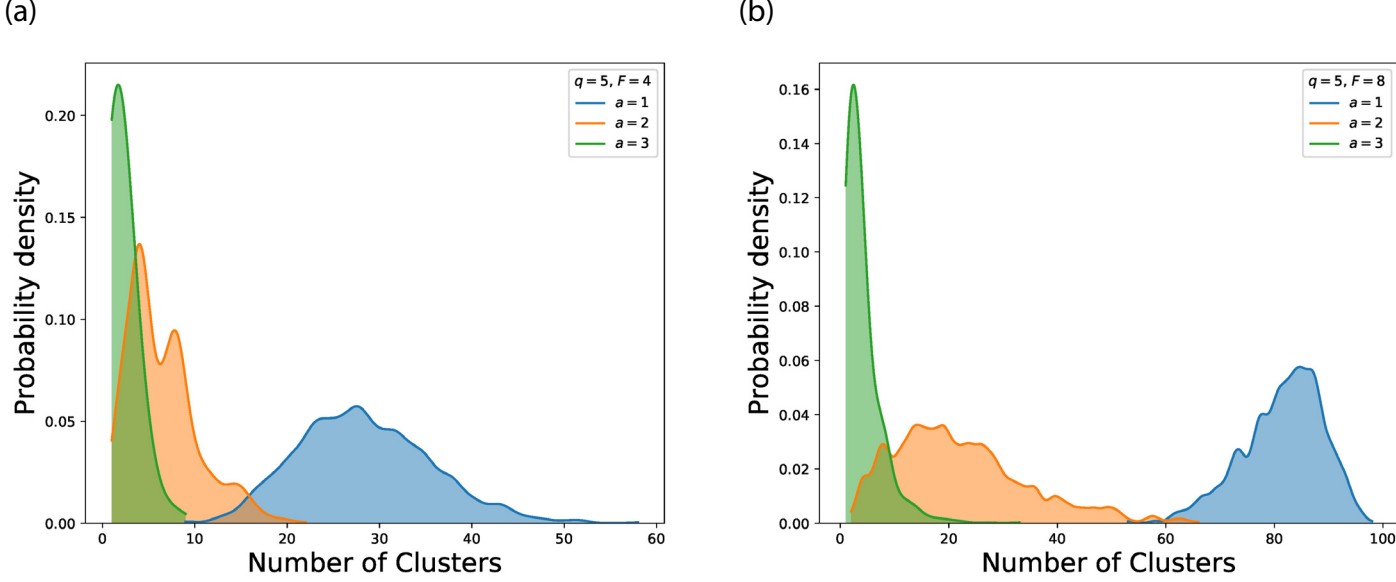

**Fig 7. The probability density showing the distribution of the number of clusters for 1,000 simulations with *q* = 5 and varying the agreement threshold *a*.** The figure on the left has *F* = 4 on the right *F* = 8. As the number of features increases, larger clusters are harder to form as there is a much larger feature-space to agree on.

(a)

(b)

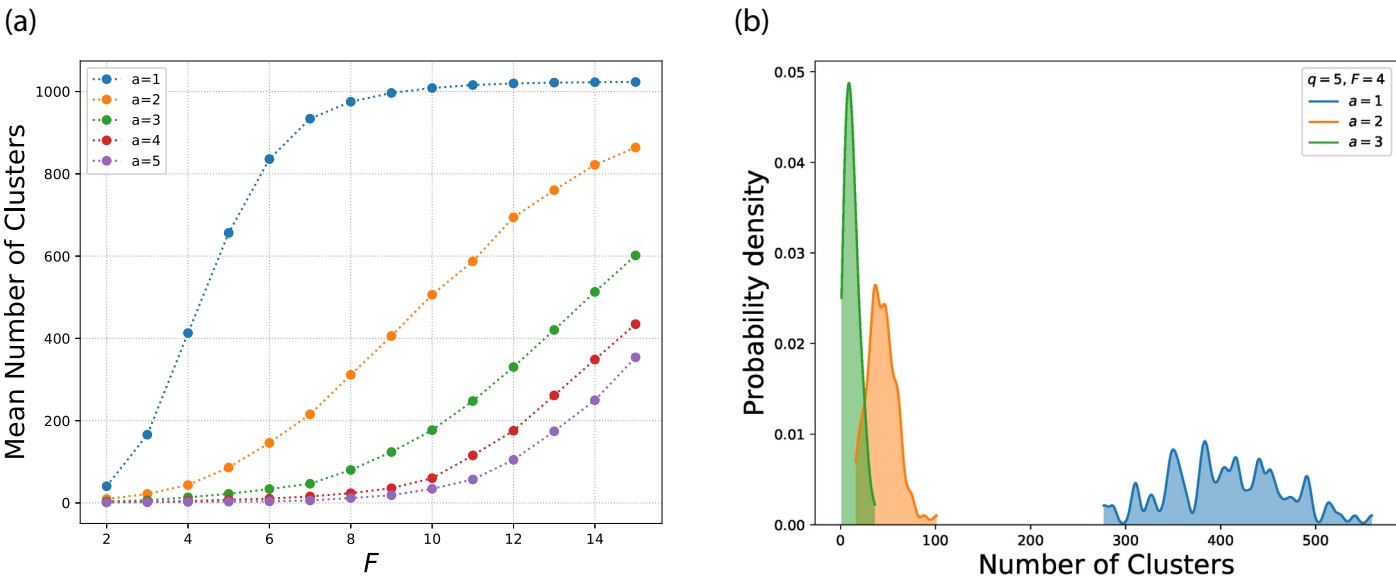

**Fig 8. The mean number of clusters per feature for $q = 7$ (left) and the probability density for the number of clusters for $q = 7$ and $F = 4$ (right) for a system with $N = 1024$.** The behaviour is similar for larger systems.

We expect that it will be relatively straightforward to make similar modifications to other opinion dynamics models to allow the consideration of survey-type data.

The code to run bipartite simulations and the agreement threshold can be found here: https://github.com/pmaccarron/Agreement-Threshold-Axelrod.

## Acknowledgments

The authors would like to thank the attendees of the workshop "ToRealSim meets DAFINET: Linking agent based models with empirical evidence", 4-6 April 2019, University of Limerick, Ireland. In particular the authors thank Michael Mäs, Sylvie Huet and Guillaume Deffuant for their detailed feedback on this manuscript.

## Author Contributions

**Conceptualization:** Pádraig MacCarron, Paul J. Maher, Susan Fennell, James P. Gleeson, Kevin Durrheim, Michael Quayle.

**Formal analysis:** Pádraig MacCarron.

**Funding acquisition:** James P. Gleeson, Michael Quayle.

**Methodology:** Pádraig MacCarron, Kevin Burke, Michael Quayle.

**Software:** Pádraig MacCarron.

**Supervision:** Michael Quayle.

**Visualization:** Pádraig MacCarron.

**Writing – original draft:** Pádraig MacCarron.

**Writing – review & editing:** Pádraig MacCarron, Paul J. Maher, Susan Fennell, Kevin Burke, James P. Gleeson, Michael Quayle.

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
