## [Decision Letter · Decision Letter 0]

17 Dec 2019

PONE-D-19-31281

Agreement Threshold on Axelrod's model of Cultural Dissemination

PLOS ONE

Dear Dr. Mac Carron,

Thank you for submitting your manuscript to PLOS ONE. After careful consideration, we feel that it has merit but does not meet PLOS ONE’s publication criteria as it currently stands. Therefore, we invite you to submit a revised version of the manuscript that addresses the points raised during the review process.

We would appreciate receiving your revised manuscript by Jan 26 2020 11:59PM. To enhance the reproducibility of your results, we recommend that if applicable you deposit your laboratory protocols in protocols.io, where a protocol can be assigned its own identifier (DOI) such that it can be cited independently in the future. For instructions see: http://journals.plos.org/plosone/s/submission-guidelines#loc-laboratory-protocols

We look forward to receiving your revised manuscript.

Kind regards,

Carlos Gracia-Lázaro

Academic Editor

PLOS ONE

**When submitting your revision, we need you to address these additional requirements:**

**Please ensure that your manuscript meets PLOS ONE's style requirements, including those for file naming. The PLOS ONE style templates can be found at http://www.plosone.org/attachments/PLOSOne_formatting_sample_main_body.pdf and http://www.plosone.org/attachments/PLOSOne_formatting_sample_title_authors_affiliations.pdf**

Additional Editor Comments (if provided):

The authors should pay attention to the reviewers' suggestions inviting to a deep revision of the manuscript both in its form and in its content and methods.

Also, the authors should pay attention to the journal's criteria of code availability.

Reviewers' comments:

Reviewer's Responses to Questions

**Comments to the Author**

1. Is the manuscript technically sound, and do the data support the conclusions?

Reviewer #1: Yes

Reviewer #2: Partly

Reviewer #3: No

2. Has the statistical analysis been performed appropriately and rigorously? 

Reviewer #1: No

Reviewer #2: N/A

Reviewer #3: No

3. Have the authors made all data underlying the findings in their manuscript fully available?

Reviewer #1: Yes

Reviewer #2: No

Reviewer #3: Yes

4. Is the manuscript presented in an intelligible fashion and written in standard English?

Reviewer #1: Yes

Reviewer #2: Yes

Reviewer #3: Yes

5. Review Comments to the Author

Reviewer #1: PONE-D-19-31281

Agreement Threshold on Axelrod's model of Cultural Dissemination

The manuscript contains two distinct contributions regarding the well-known Axelrod model of cultural dissemination. First, a visualization technique based on representing the system as a bipartite graph where the two node types are agents and traits (specifically one node for each trait value in each feature so there are qF such nodes using the usual notation of F for the feature vector dimension and q for the number of trait values). The one-mode projections of this graph can then be used to show connections between attitudes (where nodes are the traits) or connections between agents with common attitudes (where nodes are the agents).

The second aspect of the manuscript is the introduction of an "agreement threshold" based on the social judgment theory concept of "latitude of acceptance". The Axelrod model is modified so that the probability of interaction is no longer proportional simply to cultural similarity (Hamming distance between culture vectors) but instead proportional to the number of features within the agreement threshold a. (Note that this depends on the traits now being treated as ordinal rather than categorical [nominal]). Further, rather than any randomly chosen trait (not already equal) being copied, only a trait within the agreement threshold is copied.

Although the manuscript is a potentially useful contribution, I think points 1, 2, and 9 below need to be addressed before it could published (the other points are more minor, or not a barrier to publication PLOS ONE).

1. I think there is insufficient consideration of the existing literature, which is rather extensive for the Axelrod model and various enhancements or modifications of it. There are some reviews of this literature, and two are cited (citations [28] and [29], Flache et al. (2017) and Castallano et al. (2009)), but these reviews are not necessarily exhaustive of the (very extensive) literature on the Axelrod model (and the longer review article is 10 years old).

In particular, although some of the Flache & Macy arXiv preprints introducing it are cited, I would have expected more discussion of "bounded confidence" as it is so closely related to the "agreement threshold" idea, and in particular, De Sanctis & Galla (2009) and Hegselmann & Krause (2002) are good sources for this (the latter is not about the Axelrod model, but gives some citations for the history of the "bounded confidence" idea in other models). Another very useful paper on the Axelrod model is Flache & Macy (2011), who state that:

"The bounded confidence models showed that global diversity does not depend on the assumption that opinions are discrete, as long as influence can only occur between individuals who are sufficiently similar. Thus, while Axelrod introduced two innovations—discrete opinions and homophily — these subsequent studies showed that the latter is sufficient for local convergence to preserve global diversity."

(Flache & Macy (2011), p. 972).

These papers are all cited in the Flache et al. (2017) review, which discusses bounded confidence, but I think it is necessary to do more than just cite the review in this manuscript, and actually also directly cite the individual papers and discuss bounded confidence and its relationship to the "agreement threshold".

These omissions can be easily rectified, but in such an extensively studied topic as the Axelrod model, with many variations and in different literatures (political science, physics, social psychology, etc.) it is easy to overlook relevant work and inadvertently re-invent something or fail to cite directly relevant prior work, so I believe it is particularly important to be thorough in this area.

This has particular relevance to my next point, as I think the highly relevant paper Valori et al. (2012) needs to be considered.

2. The main motivation is to use survey data with ordinal scales (e.g. Likert scales) in the Axelrod model. However (related to point 1 above) this has already been done in published literature, which is not cited in the manuscript. Specifically, Valori et al. (2012) use an Axelrod model with empirical data from surveys as the initial culture vectors (rather than the usual random initialization), as do some subsequent works, for example Babeanu et al. (2017) [see Appendix A for details of treatment of survey data with nominal and ordinal cultural features]; Babeanu et al. (2018); Stivala et al. (2014).

The details of this in the Valori et al. (2012) paper are largely in the Supplementary Information, but they are there (see S.I. p. 3), showing how both nominal and ordinal features are treated in their version of the Axelrod model. Even some of the "further work" in the manuscript such as "vary q per feature" is already implicitly described there.

In addition, the Valori et al. (2012) S.I. discusses the arbitrariness of the number of features F and how bounded confidence can compensate for this (see SI. pp. 6-7 and footnote on p. 7), which was a motivation in the manuscript for introducing the agreement threshold.

3. Similarly, although I am not familiar with any previous publications using the bipartite graph idea, the one-mode projection where nodes are agents would appear to be a weighted version of the "cultural graph" described by Valori et al. (2012), in which an edge exists between agents with sufficiently similar (according to the value of the confidence threshold) culture vectors. Although in Valori et al. (2012) and subsequent work building on it the "cultural graph" was not used for visualization, it nevertheless was defined and used for analysis of the model.

4. I am not convinced of the utility of the bipartite graph visualization (e.g. Fig. 2). It seems to me that the presence of the agents as nodes and the density of lines showing the feature-trait combinations really just clutters the visualization (and would quickly become unreadable for larger values of N, q, and F). Would it not be simpler just to represent the relative proportions of the feature-trait combination as a heat map, for example? Valori et al. (2012) visualize correlations between opinions as heat maps and dendrograms (as they are interested in the structure of such correlations), but the simple proportions of each trait could also be visualized directly this way.

5. Although I believe the "agreement threshold" is a new and useful contribution, I found its explanation a little confusing at first. I think it might be clearer to specify it not just in English but also precisely in mathematical notation, as is done in the physics literature on the Axelrod model for describing the measure of cultural similarity etc. In the abstract (where mathematical notation is inappropriate) on a first reading it seemed to me that the "agreement threshold" was just the same as "bounded confidence", and it required reading the manuscript carefully to see that it is not.

6. On p. 2 "Specifically, when the number of features F is greater than the number of traits q , usually there will be consensus with only one cluster emerging [8]". I'm not sure that this statement is supported by citation [8] (Castellano et al. (2000)). [See point (2) above for more on how the arbitrariness of the number of features F has previously been handled for survey data using bounded confidence.]

7. Why is the number of clusters counted with the "dispersion indicator" rather than a more conventional order parameter for the Axelrod model such as the mean size of the largest cluster, or simply the number of cultures? There is a mention of phase transitions as further work in the Conclusions section, and indeed no phase transition is apparent in Fig. 6. Presumably this is because the quantity on the x axis does not control a phase transition (while q does, for example). Or perhaps is the "dispersion indicator" actually not an order parameter?

8. I found Figure 6 (or more precisely its axis labeling) confusing. What is the x axis exactly? It is labeled "No. features" so it seems to be the dimension of the culture vector i.e. F. If so I think it would be better to clearly state this. [There is also a typo in the caption, should be q = 7 not q - 7].

9. I don't think the computational experiments are thorough enough. In the absence of any analytical treatment (mean-field approximation for example) simulation experiments are all that can be done, which is reasonable. However the only computational experiments seem to be those represented in Figure 6 showing the dispersion indicator in the absorbing state (final state) for two values of q (and the corresponding possible values of a) and one value of N. There are no error bars so it would seem these represent just a single run of the model for each parameter combination. At a minimum I would expect the results of multiple runs (from random initial conditions) and the figure then showing the mean and standard deviation (or 95% confidence interval) over many runs, as is done in the vast majority of publications on Axelrod model variants.

10. The manuscript defines a bipartite network as "a network with two types of nodes" (p. 2) but omits the essential point that any edge must be between nodes of two different types (never two nodes of the same type). Indeed a bipartite graph is conventionally defined as a graph in which the nodes can be partitioned into two disjoint sets such that every edge connects a node in one set to a node in the other.

11. Although PLOS ONE requires data be made freely available, it appears to have no such requirement for code. Hence I cannot insist that the code for the modified Axelrod model with agreement threshold be made available, nevertheless I think it would be preferable to do so to facilitate reproducbility. Even a "toy" version in e.g. NetLogo would allow people to verify their undertanding of the model description in the manuscript (see also point (5) above) or compare their own implementation to check they match.

References (in addition to those cited in the manuscript)

Băbeanu, A. I., Talman, L., & Garlaschelli, D. (2017). Signs of universality in the structure of culture. The European Physical Journal B, 90(12), 237.

Băbeanu, A. I., van de Vis, J., & Garlaschelli, D. (2018). Ultrametricity increases the predictability of cultural dynamics. New Journal of Physics, 20(10), 103026.

De Sanctis, L., & Galla, T. (2009). Effects of noise and confidence thresholds in nominal and metric Axelrod dynamics of social influence. Physical Review E, 79(4), 046108.

Flache, A., & Macy, M. W. (2011). Local convergence and global diversity: From interpersonal to social influence. Journal of Conflict Resolution, 55(6), 970-995.

Hegselmann, R., & Krause, U. (2002). Opinion dynamics and bounded confidence models, analysis, and simulation. Journal of artificial societies and social simulation, 5(3).

Stivala, A., Robins, G., Kashima, Y., & Kirley, M. (2014). Ultrametric distribution of culture vectors in an extended Axelrod model of cultural dissemination. Scientific reports, 4, 4870.

Valori, L., Picciolo, F., Allansdottir, A., & Garlaschelli, D. (2012). Reconciling long-term cultural diversity and short-term collective social behavior. Proceedings of the National Academy of Sciences, 109(4), 1068-1073.

Reviewer #2: Please see attached review report.....................................................................................................................................................................................................................................................................................

Reviewer #3: In this paper, the authors propose modifications to the Axelrod model. This work supposedly has two contributions: a method for displaying the system state through a bipartite network, and a modification in the step which defines how agents interact.

It is unclear if the method for displaying the system state as a bipartite graph provides any benefit. For instance, the conclusions they reach through Figure 2 would be easier to observe through a usual frequency plot. Furthermore, the figures provide a poor visualization of the bipartite graph and it is difficult to see where are the traits nodes. If there was some benefit in the bipartite representation they should make use of this representation to achieve something that would not be possible without a network, e.g., finding communities or computing distances. This is not the case in this paper.

Lastly, their modification to the Axelrod model is not explored seriously. Results are analysed in terms of individual executions without any clear demonstration of how representative they are. Fig. 6 seems to be the only one corresponding to ensemble averages, however, it lacks statistical information. As they posed in the conclusion, it is necessary to analyse the change of behaviour for different values of F, q, a, and N. Furthermore, their most remarkable conclusion is concerning the different expectations for the number of clusters in the final states. However, according to the authors model, when agents interact, only the features that have traits within the agreement threshold can be copied. This behaviour can make some features never being copied, which would lead to a final state without consensus. Therefore, their finding of "a large number of clusters" might be an effect of this limitation and not the agreement threshold.

Specific remarks

- Fig. 1 and 2 supposedly correspond to a simulation with F=3 and q=5. However, the authors explain the visualization for F=5.

- Authors refer to the paper for attitudes and response-options, but it is not clear how they are related to features or traits. Do attitudes correspond to the number of traits or features?

- To verify the number of clusters in the simulation, the authors use the dispersion indicator. However, if the goal is to count the number of clusters, wouldn't be more natural to check directly the number of clusters?

- Some phrases of the manuscript are really hard to understand and the authors should work on the writing, e.g., line 180.

6. PLOS authors have the option to publish the peer review history of their article (what does this mean?). If published, this will include your full peer review and any attached files.

Reviewer #1: No

Reviewer #2: No

Reviewer #3: No

---

## [Author Response · Author response to Decision Letter 0]

27 Mar 2020

Reviewer #1

1. I think there is insufficient consideration of the existing literature, which is rather extensive for the Axelrod model and various enhancements or modifications of it. There are some reviews of this literature, and two are cited (citations [28] and [29], Flache et al. (2017) and Castallano et al. (2009)), but these reviews are not necessarily exhaustive of the (very extensive) literature on the Axelrod model (and the longer review article is 10 years old).

Reply: Thanks for this, we have added more references and expanded the introduction (in red) and are happy to add more or expand more on this if necessary.

In particular, although some of the Flache & Macy arXiv preprints introducing it are cited, I would have expected more discussion of "bounded confidence" as it is so closely related to the "agreement threshold" idea, and in particular, De Sanctis & Galla (2009) and Hegselmann & Krause (2002) are good sources for this (the latter is not about the Axelrod model, but gives some citations for the history of the "bounded confidence" idea in other models). Another very useful paper on the Axelrod model is Flache & Macy (2011), who state that:

"The bounded confidence models showed that global diversity does not depend on the assumption that opinions are discrete, as long as influence can only occur between individuals who are sufficiently similar. Thus, while Axelrod introduced two innovations—discrete opinions and homophily — these subsequent studies showed that the latter is sufficient for local convergence to preserve global diversity."

(Flache & Macy (2011), p. 972).

These papers are all cited in the Flache et al. (2017) review, which discusses bounded confidence, but I think it is necessary to do more than just cite the review in this manuscript, and actually also directly cite the individual papers and discuss bounded confidence and its relationship to the "agreement threshold".

Reply: Bounded confidence was the main reason we came up with this model so was a big influence, however, we removed this after multiple discussions from those mentioned in the acknowledgements. Their argument was that with the traits and bound not being continuous it was too confusing to call it bounded confidence when we were dealing with discrete values – even though there are models out there that take discrete values. Also you’ll see at one point we discuss something about not using continuous values, the survey discussion was added in after to convince them that taking discrete values had merit. Initially this paper was framed more theoretically in terms of adding this bounded confidence extension (which we later renamed an agreement threshold), we added the survey framing after their comments. 

2. The main motivation is to use survey data with ordinal scales (e.g. Likert scales) in the Axelrod model. However (related to point 1 above) this has already been done in published literature, which is not cited in the manuscript. Specifically, Valori et al. (2012) use an Axelrod model with empirical data from surveys as the initial culture vectors (rather than the usual random initialization), as do some subsequent works, for example Babeanu et al. (2017) [see Appendix A for details of treatment of survey data with nominal and ordinal cultural features]; Babeanu et al. (2018); Stivala et al. (2014).

The details of this in the Valori et al. (2012) paper are largely in the Supplementary Information, but they are there (see S.I. p. 3), showing how both nominal and ordinal features are treated in their version of the Axelrod model. Even some of the "further work" in the manuscript such as "vary q per feature" is already implicitly described there.

In addition, the Valori et al. (2012) S.I. discusses the arbitrariness of the number of features F and how bounded confidence can compensate for this (see SI. pp. 6-7 and footnote on p. 7), which was a motivation in the manuscript for introducing the agreement threshold.

Reply: Many thanks for this, though familiar with the paper I confess I had not looked at the SI. We have added this to the introduction now and again are happy to elaborate further if necessary.

3. Similarly, although I am not familiar with any previous publications using the bipartite graph idea, the one-mode projection where nodes are agents would appear to be a weighted version of the "cultural graph" described by Valori et al. (2012), in which an edge exists between agents with sufficiently similar (according to the value of the confidence threshold) culture vectors. Although in Valori et al. (2012) and subsequent work building on it the "cultural graph" was not used for visualization, it nevertheless was defined and used for analysis of the model.

Reply: We have added a sentence on this.

4. I am not convinced of the utility of the bipartite graph visualization (e.g. Fig. 2). It seems to me that the presence of the agents as nodes and the density of lines showing the feature-trait combinations really just clutters the visualization (and would quickly become unreadable for larger values of N, q, and F). Would it not be simpler just to represent the relative proportions of the feature-trait combination as a heat map, for example? Valori et al. (2012) visualize correlations between opinions as heat maps and dendrograms (as they are interested in the structure of such correlations), but the simple proportions of each trait could also be visualized directly this way.

Reply: For pure Axelrod as the feature-trait combination are arbitrary it can look strange, however increasing in size won’t be a problem as the brightness increases. However, out main interest in this type of visualisation will come when we actually apply it to survey data, see this image for example:

https://www.dropbox.com/s/d8zv783gf4ijy19/attitudes_net_2113_kk.png?dl=0

The two types of nodes are people and their responses to eight questions, as the features (questions) and traits (responses) are no longer arbitrary we think it’s a nice way to visualise this type of data which is why we included it here. We decided we wanted to keep this paper theoretical though so did not add the empirical data.

5. Although I believe the "agreement threshold" is a new and useful contribution, I found its explanation a little confusing at first. I think it might be clearer to specify it not just in English but also precisely in mathematical notation, as is done in the physics literature on the Axelrod model for describing the measure of cultural similarity etc. In the abstract (where mathematical notation is inappropriate) on a first reading it seemed to me that the "agreement threshold" was just the same as "bounded confidence", and it required reading the manuscript carefully to see that it is not.

Reply: We have added an equation to the section now.

6. On p. 2 "Specifically, when the number of features F is greater than the number of traits q , usually there will be consensus with only one cluster emerging [8]". I'm not sure that this statement is supported by citation [8] (Castellano et al. (2000)). [See point (2) above for more on how the arbitrariness of the number of features F has previously been handled for survey data using bounded confidence.]

Reply: Perhaps this is better phrased as multicultularity is found for very high numbers of q are required but in out work we have a small value (5 or 7 typically). We have edited this and added that bounded confidence was suggested as a fix for this.

7. Why is the number of clusters counted with the "dispersion indicator" rather than a more conventional order parameter for the Axelrod model such as the mean size of the largest cluster, or simply the number of cultures? 

Reply: We initially used cluster size but were recommended the dispersion indicator, however we are happy to go back to number of clusters. We went with the mean (of 1000 simulations) and rather than use standard deviation or the 5 and 95 percentile, we instead show another figure with the density plot of a specific q and F varying the agreement threshold to get an idea of effect of the changing parameters and the distributions are clearer to see than with error bars in the plot for the means. (Note we have done it with the error bars but feel there is a bit too much going on, see:

https://www.dropbox.com/s/c2x1gb5x2nj3phs/n_clusters_95_percentile.png?dl=0

however, we are happy to put this in also if necessary).

There is a mention of phase transitions as further work in the Conclusions section, and indeed no phase transition is apparent in Fig. 6. Presumably this is because the quantity on the x axis does not control a phase transition (while q does, for example). Or perhaps is the "dispersion indicator" actually not an order parameter?

Reply: We removed the dispersion index as, although we were recommended to use it, we didn’t really understand it clearly. We also removed mention of a phase transition.

8. I found Figure 6 (or more precisely its axis labeling) confusing. What is the x axis exactly? It is labeled "No. features" so it seems to be the dimension of the culture vector i.e. F. If so I think it would be better to clearly state this. [There is also a typo in the caption, should be q = 7 not q - 7].

Reply: Thanks, we have edited this now.

9. I don't think the computational experiments are thorough enough. In the absence of any analytical treatment (mean-field approximation for example) simulation experiments are all that can be done, which is reasonable. However the only computational experiments seem to be those represented in Figure 6 showing the dispersion indicator in the absorbing state (final state) for two values of q (and the corresponding possible values of a) and one value of N. There are no error bars so it would seem these represent just a single run of the model for each parameter combination. At a minimum I would expect the results of multiple runs (from random initial conditions) and the figure then showing the mean and standard deviation (or 95% confidence interval) over many runs, as is done in the vast majority of publications on Axelrod model variants.

Reply: We have changed to the mean number of clusters and provide the density plots as well as do this for both N = 100 and N = 1024 now. Also we have made it more clear that each point represents 1,000 simulations and expanded on the discussion in that section.

10. The manuscript defines a bipartite network as "a network with two types of nodes" (p. 2) but omits the essential point that any edge must be between nodes of two different types (never two nodes of the same type). Indeed a bipartite graph is conventionally defined as a graph in which the nodes can be partitioned into two disjoint sets such that every edge connects a node in one set to a node in the other.

Reply: We have put in that edges have to connect different types of nodes now to clarify this.

11. Although PLOS ONE requires data be made freely available, it appears to have no such requirement for code. Hence I cannot insist that the code for the modified Axelrod model with agreement threshold be made available, nevertheless I think it would be preferable to do so to facilitate reproducbility. Even a "toy" version in e.g. NetLogo would allow people to verify their undertanding of the model description in the manuscript (see also point (5) above) or compare their own implementation to check they match.

Reply: The code was written in Python so I have uploaded a script to my github and linked at the end.

Reviewer #2

1) The code to run the simulations and to produce the novel plots presented in the

manuscript is not made available. No one can evaluate the results of the model without the

code. Importantly, the new plotting techniques are a potentially valuable contribution of the

manuscript. However readers interested in using these methods are forced to figure out how

to implement them on their own. I would recommend writing the ABM and the plotting

scripts in R, as this would make these innovations accessible to a broad range of researchers

from different disciplines.

Reply: The code was written in Python so I have uploaded a script to my github and put the link at the end.

I found Figure 1 to be confusing. Because nodes are connected with lines representing

their similarity, it was difficult for me to discern the boundaries between clusters. Instead,

I would recommend using lines in the spaces around nodes (rather than lines connecting

nodes) to represent similarity/difference, and presenting a time series of plots so that read-

ers can see the clusters develop over time. In other words, model your figure after the original

Axelrod (1997) Figure 1.

Reply: The reason we chose to do it this way is on the next figure we want to represent a node as an agent in the bipartite graph. Then there is a connection between that node and its attitude, if we use the “site” which is what is traditionally done then it is hard to translate that to a node when connecting it to the attitude. We have put in a statement on this hopefully making it less confusing.

3) I don’t understand the interpretation given for the dispersion parameter d in equation

1. It seems that d is equivalent to n in equation 9 of the cited reference (Duffuant 2006),

which is the “generalized number of clusters” according to this reference. However, in the

manuscript it is stated that the inverse of d represents the generalized number of clusters

(lines 154-155). Similarly, it is stated that Figure 6 plots d, which does seem to be the inverse

of the number of clusters. Am I correct that what you have actually plotted in Figure 6 is

1/d?

Reply: We have changed to the mean number of clusters and provide the density plots as well as do this for both N = 100 and N = 1024 now. Also we have made it more clear that each point represents 1,000 simulations and expanded on the discussion in that section. We hope that this has made that section clearer.

4) I don’t understand the explanation for why adding an agreement threshold a results in

an increasing number of clusters at equilibrium as the number of features (F ) increases. The

explanation given on lines 160-162 is: “As the number of features increases, there are a larger

number of feature-trait combinations an agent can have. For a low agreement threshold an

agent would not be able to interact with any of its neighbours.” If I understand correctly

what is written, this process serves to increase the number of isolated nodes at equilibrium,

which in turn contributes to a larger number of clusters. However, this runs counter to

my intuition. It’s quite possible that I have misunderstood something about the model, so

I will outline my thinking below. If there is some error in my logic, perhaps the authors

would consider including some additional explanation in the paper so that readers don’t

make similar mistakes.

Reply: The agreement threshold is used in the copying of traits as well as in the calculation of a probability for interaction.

The probability for interaction between two nodes is increased when the agreement threshold is included (vs standard Axelrod) as features in which you are (for example) 1 trait away from each other now add to the probability for interaction. However, once you decide interaction can occur, the only features you can change are the ones in which the traits are sufficiently similar (within the agreement threshold). In standard Axelrod once interaction happens one feature will be copied, thereby increasing the probability of interaction in the future. In our version you only change the features which are already quite similar, meaning the probability for interaction in the future is the same.

We have added something on this hopefully clarifying it further but are happy to add more if necessary.

Reviewer #3

It is unclear if the method for displaying the system state as a bipartite graph provides any benefit. For instance, the conclusions they reach through Figure 2 would be easier to observe through a usual frequency plot. Furthermore, the figures provide a poor visualization of the bipartite graph and it is difficult to see where are the traits nodes. If there was some benefit in the bipartite representation they should make use of this representation to achieve something that would not be possible without a network, e.g., finding communities or computing distances. This is not the case in this paper.

Lastly, their modification to the Axelrod model is not explored seriously. Results are analysed in terms of individual executions without any clear demonstration of how representative they are. Fig. 6 seems to be the only one corresponding to ensemble averages, however, it lacks statistical information. 

Reply: We have changed to the mean number of clusters and provide the density plots as well as do this for both N = 100 and N = 1024 now. Also we have made it more clear that each point represents 1,000 simulations and expanded on the discussion in that section.

Specific remarks

- Authors refer to the paper for attitudes and response-options, but it is not clear how they are related to features or traits. Do attitudes correspond to the number of traits or features?

Reply: We mean the combination, if on a survey your response to question 1 is: 3, then your attitude is question1-response3. 

- To verify the number of clusters in the simulation, the authors use the dispersion indicator. However, if the goal is to count the number of clusters, wouldn't be more natural to check directly the number of clusters?

Reply: We have changed to the number of clusters and provide the density plot to visualise the distribution.

- Some phrases of the manuscript are really hard to understand and the authors should work on the writing, e.g., line 180.

Reply: Line 180 for us is in the acknowledgements, however we have cleared up many sentences as highlighted in the text.

---

## [Decision Letter · Decision Letter 1]

12 Apr 2020

PONE-D-19-31281R1

Agreement Threshold on Axelrod's model of Cultural Dissemination

PLOS ONE

Dear Dr. Mac Carron,

Thank you for submitting your manuscript to PLOS ONE. After careful consideration, we feel that it has merit but does not fully meet PLOS ONE’s publication criteria as it currently stands. Therefore, we invite you to submit a revised version of the manuscript that addresses the points raised during the review process.

We would appreciate receiving your revised manuscript by May 27 2020 11:59PM. To enhance the reproducibility of your results, we recommend that if applicable you deposit your laboratory protocols in protocols.io, where a protocol can be assigned its own identifier (DOI) such that it can be cited independently in the future. For instructions see: http://journals.plos.org/plosone/s/submission-guidelines#loc-laboratory-protocols

We look forward to receiving your revised manuscript.

Kind regards,

Carlos Gracia-Lázaro

Academic Editor

PLOS ONE

Reviewers' comments:

Reviewer's Responses to Questions

**Comments to the Author**

1. If the authors have adequately addressed your comments raised in a previous round of review and you feel that this manuscript is now acceptable for publication, you may indicate that here to bypass the “Comments to the Author” section, enter your conflict of interest statement in the “Confidential to Editor” section, and submit your "Accept" recommendation.

Reviewer #1: All comments have been addressed

Reviewer #2: (No Response)

Reviewer #3: (No Response)

2. Is the manuscript technically sound, and do the data support the conclusions?

Reviewer #1: Yes

Reviewer #2: Yes

Reviewer #3: No

3. Has the statistical analysis been performed appropriately and rigorously? 

Reviewer #1: Yes

Reviewer #2: Yes

Reviewer #3: Yes

4. Have the authors made all data underlying the findings in their manuscript fully available?

Reviewer #1: Yes

Reviewer #2: Yes

Reviewer #3: Yes

5. Is the manuscript presented in an intelligible fashion and written in standard English?

Reviewer #1: Yes

Reviewer #2: Yes

Reviewer #3: Yes

6. Review Comments to the Author

Reviewer #1: PONE-D-19-31281R1

Agreement Threshold on Axelrod's model of Cultural Dissemination

Revision 1 (after first round of review)

The authors have addressed all my concerns and the revised manuscript is now acceptable for publication, in my opinion.

In particular, noting that the results in the figures are means over 1000 runs is very important, as it did appear in the original manuscript that these were single runs. I agree that the plots with error bars (as shown in the Dropbox link in the response) are unreadable, so showing the PDF in a separate figure rather than adding error bars is better.

I now understand the authors' reasoning about the bipartite graph visualizations and this being a theoretical rather than empirical paper, but in a way it is a shame that the survey data figure (Dropbox link from response to my point 4) is not in the paper instead of Fig 2 and Fig 3 (right), as I think the layout of the survey response figure (with the respondent nodes in the centre and the opinions on the outside) is clearer, and the concreteness of the opinions ("Race relations anti" etc.) makes the meaning a lot clearer than "f1_q1" etc. Presumably this will be for a future paper, however.

I am also pleased that the code is now publicly available. I briefly tried it and it found it to work as expected (after changing the output file type as I could not get Matplotlib, which I always have problems with, to work with MPEG on my system).

However, regarding my point (10) on defining "bipartite network", the revised text (at lines 76-77, page 3) now has

"... a network with two types of nodes which cannot have edges connecting

the different types of nodes."

which to my mind is confusing or ambiguous: the essential point is that no edge can join two nodes of the same type. This new sentence, however, could be interpreted as saying just the opposite (that no edge can join two nodes of different types). So I would suggest re-phrasing this sentence to make it clearer.

Reviewer #2: For the most part, I believe my previous comments have been adequately addressed and I would now recommend this manuscript for publication. I have two minor suggestions:

1) The authors' response in the Response to Reviewers letter to Reviewer 2 point #4 was very helpful for me. I would suggest putting that response directly into the manuscript.

2) In Methods: Standard Axelrod Model, penultimate sentence in first paragraph. I think it would be clearer to say "edges must connect nodes of different types"

Reviewer #3: 1. I believe the authors have not provided an appropriate response to the comments made by myself and by reviewer 1 about the usefulness of the graph bipartite representation. According to their comment on line 93: "For example, we observe that features 1 and 2 both have a majority but feature 3 is split between q = 1 and q = 4. This is particularly useful if using this network to represent survey data. If there are three questions with a five-scale response, it is clear which response is the most favoured per question." The authors are reaching this conclusion through the degree of the nodes in the graph plot, which would be a lot clearer in a simple plot showing the frequency of traits per feature. Besides this, I don't see any benefit provided by the bipartite representation and don't believe it is a contribution of the paper. For it to be a contribution, the authors should have used it for some analysis that needed the network representation. Now, it seems to be a rather unnecessary artefact.

2. Still regarding the bipartite representation, the authors mention in the abstract: "This visualisation is particularly useful when representing survey data as it illustrates the coevolution of cultures and opinion-based groups in Axelrod’s model of cultural diffusion." In the introduction, however, authors mention: "Opinion-based groups (or “cultures”) are formed by people holding a particular selection of attitudes." It seems, thus, that cultures and opinion-based groups correspond to the same concept. What would they refer to when discussing about coevolution? Furthermore, it is unclear how this visualization illustrates the supposed coevolution as it is just a representation of the stationary state.

3. Intuitively, it makes sense the use of the agreement thresholds for computing the probability of interaction between agents. Nonetheless, its use in the process of social influence, i.e., copying traits of features that are in the agreement threshold, seems rather unnatural. The authors should add some reference justifying why social influence should only occur in features that are $a$ traits away. Furthermore, this characteristic of the model should be the motive it does not reach consensus for small q and a, as consensus can be unreachable from initialization.

4. In the conclusions, the authors mention: " The

clustering that emerges is useful for considering opinion-based groups in opinion

dynamics models and empirical data such as surveys. " It is unclear how the clusters found in this model can be useful for the applications mentioned. The authors should discuss this more thoroughly.

5. On line 192: "One outcome of this model is that extremists are less likely to change their position

than moderates." This is not an outcome of the model, this is given by the specification of the model. If agents cannot copy the features which are not in the agreement threshold, features with extreme value traits cannot be modified unless the agent's neighbours features are also in the same extreme.

Specific remarks:

- The authors update the definition of the bipartite network according to the comment of reviewer 1, however, the definition still is not precisely correct.

- What is the total number of agents, F, q in the simulations of figure 5?

- "A further extension could specify that if the agreement threshold is larger than one, the traits move towards each other rather than one agent copying the other." This actually could be used without the restraint posed by the initial condition of the model, i.e., of only copying features which are inside the agreement threshold. In my opinion, this makes more sense than the current approach.

7. PLOS authors have the option to publish the peer review history of their article (what does this mean?). If published, this will include your full peer review and any attached files.

Reviewer #1: No

Reviewer #2: No

Reviewer #3: No

---

## [Author Response · Author response to Decision Letter 1]

23 Apr 2020

Reviewer #3: 1. I believe the authors have not provided an appropriate response to the comments made by myself and by reviewer 1 about the usefulness of the graph bipartite representation. According to their comment on line 93: "For example, we observe that features 1 and 2 both have a majority but feature 3 is split between q = 1 and q = 4. This is particularly useful if using this network to represent survey data. If there are three questions with a five-scale response, it is clear which response is the most favoured per question." The authors are reaching this conclusion through the degree of the nodes in the graph plot, which would be a lot clearer in a simple plot showing the frequency of traits per feature. Besides this, I don't see any benefit provided by the bipartite representation and don't believe it is a contribution of the paper. For it to be a contribution, the authors should have used it for some analysis that needed the network representation. Now, it seems to be a rather unnecessary artefact.

Reply: We have added in a paragraph about taking projections from the bipartite representation which is what the figures with clusters represent to show that this is relevant later, for example figures 4 and 5 are projections from the bipartite graph in figure 3. This is subtly different to cultures in standard Axelrod (see response to point 2 for more details) as here a cluster represents groups of people who fully agree, they do not need to be separated. In figure 1 if the lower left and upper right regions have the same traits for each feature in standard Axelrod then they would be two cultures, using the bipartite projection, they would be one opinion-based group. We have added text to clarify this. 

We appreciate from a model point of view this can look strange, but from an empirical point of view this is a different way of viewing survey data. For example, the image we mentioned to Reviewer #1 in the first review: 

https://www.dropbox.com/s/d8zv783gf4ijy19/attitudes_net_2113_kk.png?dl=0

shows a survey with two types of nodes being the participants and their responses to eight questions, as the features (questions) and traits (responses) are no longer arbitrary we think it’s a nice way to visualise this type of data which is why we included it here. We decided we wanted to keep this paper theoretical though so did not add the empirical data.

 This is something from our upcoming empirical paper which we do not really want to include here, we would also like to use this agreement threshold model on empirical data after which is why we would rather leave this bipartite visualisation in.

2. Still regarding the bipartite representation, the authors mention in the abstract: "This visualisation is particularly useful when representing survey data as it illustrates the coevolution of cultures and opinion-based groups in Axelrod’s model of cultural diffusion." In the introduction, however, authors mention: "Opinion-based groups (or “cultures”) are formed by people holding a particular selection of attitudes." It seems, thus, that cultures and opinion-based groups correspond to the same concept. What would they refer to when discussing about coevolution? Furthermore, it is unclear how this visualization illustrates the supposed coevolution as it is just a representation of the stationary state.

Reply: This is an important point that we didn’t properly distinguish between. In Axelrod (1997), a culture in the model is a group of agents sharing the same traits for each feature that are physically next to one another. Here, however, as we take a projection from the bipartite graph, the cluster of those who agree entirely are an opinion-based group even if they are physically separated. We have made this more clear in the text by including the line:

We remove this spatial constraint when identifying clusters and instead get groups of agents sharing the same traits for each feature, we refer to these as opinion-based groups.

3. Intuitively, it makes sense the use of the agreement thresholds for computing the probability of interaction between agents. Nonetheless, its use in the process of social influence, i.e., copying traits of features that are in the agreement threshold, seems rather unnatural. The authors should add some reference justifying why social influence should only occur in features that are $a$ traits away. Furthermore, this characteristic of the model should be the motive it does not reach consensus for small q and a, as consensus can be unreachable from initialization.

Reply: Thank you for pointing this out. We had previously included references justifying this feature of social influence, that have now been re-added in the introduction. Further justification is that people tend to display bias in evaluating evidence by favouring views that correspond to their existing attitudes (Lord, ross & Lepper, 1979; Reedy, Wells & Gastil, 2014). For example, empirical research shows that people rate arguments as more compelling when they correspond to previously held attitudes (Taber & Lodge, 2006). The mechanism at play may be biased information processing (i.e. confirmation bias), rather than social influence per se. Either way, we believe it should be accounted for in models of opinion sharing, as indeed it is in many others (Deffuant et al., 2000; Hegselmann et al., 2002). A section similar to this has been added to the introduction.

Taber, C. S., & Lodge, M. (2006). Motivated skepticism in the evaluation of political beliefs. American Journal of Political Science, 50 (3), 755–769.

Reedy, J., Wells, C. and Gastil, J. (2014), How Voters Become Misinformed: An Investigation of the Emergence and Consequences of False Factual Beliefs*. Social Science Quarterly, 95: 1399-1418. doi:10.1111/ssqu.12102

Lord, C. G., Ross, L., & Lepper, M. R. (1979) Biased assimilation and attitude polarization: The effects of prior theories on subsequently considered evidence. Journal of Personality and Social Psychology, 37(11), 2098–2109. https://doi.org/10.1037/0022-3514.37.11.2098

Deffuant G, Neau D, Amblard F, Weisbuch G (2000), Mixing beliefs among interacting

agents. Advances in Complex Systems. 01n04), 7–98.

Hegselmann R, Krause U, et al. (2002), Opinion dynamics and bounded confidence

models, analysis, and simulation. Journal of artificial societies and social

simulation. 5,3.

4. In the conclusions, the authors mention: " The clustering that emerges is useful for considering opinion-based groups in opinion dynamics models and empirical data such as surveys. " It is unclear how the clusters found in this model can be useful for the applications mentioned. The authors should discuss this more thoroughly.

Thank you for identifying this weakness in our explanation. We have strengthened the introduction in two ways. 1) We have made it clearer that the Axelrod model is a non-parametric opinion model easily adapted to ordinal data such as Likert-type scale responses and that this approach has important advantages over alternatives (e.g. averaging models). 2) We clarify that monocultural states are almost never seen in survey data (and such lack of variability is actively avoided when designing survey questions). For this reason the vanilla Axelrod model is not at all useful for modelling survey data, since as soon as the number of features (ie. survey items) exceeds the number of traits (ie. response options) homogeneity is guaranteed in a fully connected topology. 

Specifically: 

We have edited this sentence in the introduction, adding the highlighted text: “In this paper we demonstrate that an adapted version of Axelrod’s model of cultural dissemination [7] can be used to model opinion-based groups with data structures similar to survey data.”

We have edited the following sentence and added two following sentences, adding the highlighted text: “In principle, conceptualising Axelrod’s nominal cultural features and traits as ordinal attitudes will allow us to model the emergence of opinion-based groups with a data structure that maps cleanly on to raw survey data.” Note that opinion surveys typically use ordinal Likert-type response items (e.g. an item with several response options from Strongly Disagree to Strongly Agree). While these are frequently treated as interval data in analyses (ie. assuming consistent intervals between scale points and allowing arithmetic operations such as addition and subtraction), there is a strong argument that individual Likert-type items should properly be treated as ordinal, allowing only non-arithmetic operations. The original Axelrod model of cultural dissemination relies only on swapping, and our adaptions adds ranking, making it an excellent fit with Likert-type data. 

 We have changed the following sentence: “The clustering that emerges is useful for considering opinion-based groups in opinion dynamics models and empirical data such as surveys.”

Instead we argue: 

 "The resulting model is suited to natively modelling survey data, which frequently consists of Likert-type responses, since only ranking and swapping operations are required thus respecting the ordinal nature of the data. The model also avoids the homogoneous end-state which would always result when the original Axelrod model is used on survey-type data, as there are almost always features than traits (in other words, more items than response options).”

5. On line 192: "One outcome of this model is that extremists are less likely to change their position than moderates." This is not an outcome of the model, this is given by the specification of the model. If agents cannot copy the features which are not in the agreement threshold, features with extreme value traits cannot be modified unless the agent's neighbours features are also in the same extreme.

Reply: Indeed this is a specification rather than an outcome, and we have rephrased this sentence accordingly.

Specific remarks:

- The authors update the definition of the bipartite network according to the comment of reviewer 1, however, the definition still is not precisely correct.

Reply: Changed to “a graph with two types of nodes where edges must connect nodes of different types”

- What is the total number of agents, F, q in the simulations of figure 5?

Reply: F=6 and q=3, this is one of the two projections of the of figure 3 (this has been added to the caption).

- "A further extension could specify that if the agreement threshold is larger than one, the traits move towards each other rather than one agent copying the other." This actually could be used without the restraint posed by the initial condition of the model, i.e., of only copying features which are inside the agreement threshold. In my opinion, this makes more sense than the current approach.

Reply: Our model preserves those at the extremes whereas a model that allows both people move towards each other finds everyone ending up in the middle or the spectrum (See ref 36. Flache et al 2017 table 1). This prevalence of those on extremes is something that is observed in real systems (Brandt at al., 2015), we have added this reference now. 

Brandt at al (2015). The Unthinking or Confident Extremist? Political Extremists Are More Likely Than Moderates to Reject Experimenter-Generated Anchors. Psychological Science, 26(2), 189–202).

---

## [Decision Letter · Decision Letter 2]

18 May 2020

Agreement Threshold on Axelrod's model of Cultural Dissemination

PONE-D-19-31281R2

Dear Dr. Mac Carron,

We are pleased to inform you that your manuscript has been judged scientifically suitable for publication and will be formally accepted for publication once it complies with all outstanding technical requirements.

With kind regards,

Carlos Gracia-Lázaro

Academic Editor

PLOS ONE

Additional Editor Comments (optional):

Reviewers' comments:

Reviewer's Responses to Questions

**Comments to the Author**

1. If the authors have adequately addressed your comments raised in a previous round of review and you feel that this manuscript is now acceptable for publication, you may indicate that here to bypass the “Comments to the Author” section, enter your conflict of interest statement in the “Confidential to Editor” section, and submit your "Accept" recommendation.

Reviewer #2: All comments have been addressed

Reviewer #3: All comments have been addressed

2. Is the manuscript technically sound, and do the data support the conclusions?

Reviewer #2: Yes

Reviewer #3: Yes

3. Has the statistical analysis been performed appropriately and rigorously? 

Reviewer #2: Yes

Reviewer #3: Yes

4. Have the authors made all data underlying the findings in their manuscript fully available?

Reviewer #2: Yes

Reviewer #3: Yes

5. Is the manuscript presented in an intelligible fashion and written in standard English?

Reviewer #2: Yes

Reviewer #3: Yes

6. Review Comments to the Author

Reviewer #2: All of my previous comments have been addressed and I believe this manuscript is ready for publication.

Reviewer #3: I believe the authors have addressed the comments concerning the justifications for their process of social influence and other specific remarks. I have to admit that I am still not convinced by the bipartite representation process, however, it is not the main contribution of the paper. The main contribution is the model and now it seems to be justified. Therefore, I would now recommend this manuscript for publication.

7. PLOS authors have the option to publish the peer review history of their article (what does this mean?). If published, this will include your full peer review and any attached files.

Reviewer #2: No

Reviewer #3: No

---

## [Editor Report · Acceptance letter]

22 May 2020

PONE-D-19-31281R2 

Agreement Threshold on Axelrod's model of Cultural Dissemination 

Dear Dr. MacCarron:

I am pleased to inform you that your manuscript has been deemed suitable for publication in PLOS ONE. Congratulations! Your manuscript is now with our production department. 

With kind regards,

on behalf of

Dr. Carlos Gracia-Lázaro 

Academic Editor

PLOS ONE